# Sustainable, Alginate-Based Sensor for Detection of *Escherichia coli* in Human Breast Milk

**DOI:** 10.3390/s20041145

**Published:** 2020-02-19

**Authors:** Nicholas Kikuchi, Margaret May, Matthew Zweber, Jerard Madamba, Craig Stephens, Unyoung Kim, Maryam Mobed-Miremadi

**Affiliations:** 1Department of Bioengineering, Santa Clara University, Santa Clara, CA 95053, USA; nkikuchi@alumni.scu.edu (N.K.); mzweber@alumni.scu.edu (M.Z.); jmadamba@scu.edu (J.M.); 2Department of Biology, Santa Clara University, Santa Clara, CA 95053, USA; cstephens@scu.edu

**Keywords:** bioassay, breast milk, alginate, optimization, sustainable, *E. coli*, pathogen detection, Taguchi method, biosensor

## Abstract

There are no existing affordable diagnostics for sensitive, rapid, and on-site detection of pathogens in milk. To this end, an on-site colorimetric-based sustainable assay has been developed and optimized using an L_16_ (5^4^) Taguchi design to obtain results in hours without PCR amplification. To determine the level of *Escherichia coli* (*E. coli*) contamination, after induction with 150 µL of breast milk, the B-Per bacterial protein extraction kit was added to a solution containing an alginate-based microcapsule assay. Within this 3 mm spherical novel sensor design, X-Gal (5-Bromo-4-Chloro-3-Indolyl β-d-Galactopyranoside) was entrapped at a concentration of 2 mg/mL. The outward diffusing X-Gal was cleaved by β-galactosidase from *E. coli* and dimerized in the solution to yield a blue color after incubation at 40 °C. Color intensity was correlated with the level of *E. coli* contamination using a categorical scale. After an 8 h incubation period, a continuous imaging scale based on intensity normalization was used to determine a binary lower limit of detection (LOD), which corresponded to 10^2^ colony forming unit per mL (CFU/mL) and above. The cost of the overall assay was estimated to be $0.81 per sample, well under the $3 benchmark for state-of-the-art immune-based test kits for pathogen detection in biofluids. Considering the reported binary LOD cutoff of 10^2^ CFU/mL and above, this proposed hydrogel-based assay is suited to meet global requirements for screening breast milk or milk for pathogenic organisms of 10^4^ CFU/mL, with a percentage of false positives to be determined in future efforts.

## 1. Introduction

Breast milk is widely considered to be the gold standard for infant nutrition, offering essential nutrients and antibodies that enhance an infant’s health outcomes [1,2]. Breastfeeding lowers the risk of infectious disease during infancy and reduces the likelihood of chronic conditions, such as asthma, during early childhood [3]. Breastfeeding has also been linked to improved cognitive development [4] and reduced infant mortality worldwide [5]. However, mothers infected with HIV or other bloodborne diseases are advised not to breastfeed [6]. Children with metabolic conditions preventing proper digestion of breast milk or an inability to latch onto their mother’s breast to feed are also unable to obtain sustenance via breast milk [7]. In addition, pumping inaccessibility due to cultural factors also constitutes a barrier to breastfeeding [8]. These obstacles have led to a global emergence of breast milk banks that increase the accessibility of uncontaminated breast milk for many mothers and their infants [9]. Stringent controls have been put in place at these milk banks to ensure the safety of milk pumped and donated to these milk banks [10]. After testing donors for bloodborne diseases, donated milk is pasteurized to reduce the count of harmful bacteria and tested by culturing to confirm the safety of the milk [10]. Measures put in place to ensure the safety of donated milk may not always ensure that levels of bacterial contaminants are within safe levels [9], as equipment, processes, and personnel can fail [10]. However, culturing bacteria for quality control is an expensive (ranging from $35 to $81 to process approximately 100–200 ounces plus labor cost [11]) and time-consuming process [10] that is not strain-specific, resulting in a high incidence of false negatives. In addition, culture-based safety criteria are not standardized globally, leading to potential variation in bacterial content from different milk banks [10]. Highlighted in Table 1 are diagnostic technologies for milk and food, their varying purposes, detection methods, prices, applications, and times required to obtain results. Happy Vitals test kits and Milkscreen strips are commercially available, while bioassays for quality assurance of dairy products, food bioassays, electrochemical DNA-based bioassays, poly(methyl methacrylate) (PMMA) “Lab-on-a-chips” for human breast milk defatting, integrated rotary microfluidic systems, and wireless antibody-free biosensors are still in research stage. 

Papers or bioactive paper assays for bacterial detection in dairy milk and orange juice exist in a simple kit that would allow untrained personnel to carry out sensitive, multiplexed detection of *Escherichia coli* (*E. coli*) in food samples [24,25]. These paper-based assays utilize sol-gel-derived silica inks placed by an ink-jet printing technique to produce colorimetric results, which can be judged by the human eye or a combination of a digital camera and image analysis software. The bioactive paper assay is based on intracellular β-galactosidase or β-glucuronidase (GUS) hydrolysis of X-Gal (5-Bromo-4-Chloro-3-Indolyl β-d-Galactopyranoside). For *E. coli* in breast milk, the lower limit of detection (LOD) was reported to be 10^5^ colony forming unit per mL (CFU/mL) [26] for a two hour assay, at a cost of $0.34. However, the use of a paper-based assay that implements a lateral flow method for bacterial lysates to interact with an entrapped substrate may also result in contaminant entrapment, suggesting that further improvements are necessary [9].

Motivated by these observations, a bioactive assay for detection of pathogens that is sensitive (capable of detecting less than 10^5^
*E. coli*), rapid (with assay development times between two and eight hours), portable (no amplification needed and associated capital costs), environmentally friendly (no solvents and biodegradable), low-cost and requiring minimal training was developed. In order to circumvent contaminant entrapment, a simple assay design based on radial diffusion of an entrapped X-Gal substrate from an alginate microcapsule was proposed. The mass transfer component of the assay was comprised of a cross-linked alginate membrane with an established molecular weight cutoff (MWCO) of 70 kDa [27]. Alginate, composed of (1,4)-linked β-d-mannuronic and (1,3)-α-l-guluronic acid residues, is a widely used biomaterial in tissue engineering and regenerative medicine [28]. The kinetic component of the assay was based on the reaction of β-galactosidase, which hydrolyzes X-Gal (MW: 408.629 Da), releasing a substituted indole (5-bromo-4-chloro-3-hydroxyindole) that spontaneously dimerized into 5,5′-dibromo-4,4′-dichloro-indigo to give an intensely blue product [29]. The final assay design was achieved by dynamic optimization using Taguchi designs. 

## 2. Materials and Methods

### 2.1. Materials 

*E. coli* strain SCU-104, a naturally occurring commensal *E. coli* isolated from an SCU student under an institutional review board-approved (IRB-approved) protocol, was used for all testing described here. SCU-104 is capable of metabolizing lactose and shows regulatory behavior with regard to the lac operon that conforms to classic models. Luria Bertani broth (LB, cat# L3152-1KG, lot# SLBS0893V) powder, N,N-Dimethylformamide (DMF, cat# D4551-250ML, lot# SHBH9171), medium-viscosity alginic acid (cat# A2033, lot# 108K1228), and β-galactosidase (Aspergillus orizae; cat# 65160-125KU, lot# SLBB6762V, activity: 10.3 U/mg) were purchased from Sigma-Aldrich (St. Louis, MI, USA). The B-PER Direct Bacterial Protein Extraction Kit (Waltham, MA) and X-Gal (5-Bromo-4-Chloro-3-Indolyl β-d-Galactopyranoside, cat# B1690, lot# 1899771) were procured from Thermo Fisher Scientific (Waltham, MA, USA). All other reagent grade chemicals were provided by the Bioengineering Department at Santa Clara University, Santa Clara, California, USA).

### 2.2. Methods

#### 2.2.1. Cell Preparation

*E. coli* strain SCU-104 was cultured in 2 mL of 2.5% (*w*/*v*) autoclaved LB broth. Optical density (OD: 600nm) was monitored hourly using a Genesis 10S UV-Vis spectrophotometer (Thermo Fisher Scientific, Waltham, MA, USA) [30]. During assay development, when the medium reached an absorbance of 0.2–0.4 (after approximately 6 h), breast milk was added to induce expression of the lac operon. Cultures were then incubated overnight in an incubator shaker at 37 °C and 220 rpm. Subsequently, log-phase cells (approximately 10^8^ cells/mL) were harvested by centrifugation at 3000 rpm for 5 min. The bacterial cell pellet was resuspended in 200 µL of 50 mM sterile sodium phosphate buffer (pH = 7.4). A range of bacterial cell concentrations (3.16 × 10^5^–1.00 × 10^7^ CFU/mL) was prepared in order to construct a scaled detection ladder for optimization experiments. For the lower LOD determination, a range of 0–1.00 × 10^8^ (CFU/mL) was used post induction. Two separate batches of bacteria but a single batch of breast milk were used for the above-stated optimization and LOD determination experiments.

#### 2.2.2. Enzyme Extraction

*E. coli* cells were lysed in order to access intracellular β-galactosidase protein according to the B-PER extraction protocol [31]. Two hundred microliters of B-PER protein extraction solution (1–5% (*v*/*v*) lysozyme, 1–5% (*v*/*v*) DNAase, and 90–98% (*v*/*v*) B-PER) was added to *E. coli*. The solutions were then vortexed and incubated at 37 °C for lysing times ranging from 10 to 25 min. A primary positive control for enzyme induction comprised of 1 mg of X-Gal powder was added to the lysed solution corresponding to the highest concentration in the bacterial ladder (1.00 × 10^7^ CFU/mL), followed by incubation at 37 °C for 20 min. A visible color change to blue was an indicator of β-galactosidase expression. 

#### 2.2.3. Hydrogel Assay

##### Microcapsule Fabrication

Shown in Figure 1 are the microcapsule fabrication steps. A 3% (*w*/*v*) solution of alginate was prepared using 0.9% (*w*/*v*) NaCl as a solvent and autoclaved. X-Gal volumes ranging from 25 to 100 µL aliquoted from a stock solution of 20 mg X-Gal/DMF (1:19, *v*/*v*) were dissolved in 1 mL of alginate and stirred for 4 h. Alginate gel assay fabrication was based on the principle of ionotropic gelation [27]. The biopolymer mixture was extruded at 1 mL/min through a standard 304 SS 18 gauge needle (Rame Hart, NJ, USA) into a 1.5% CaCl_2_ solution for cross-linking. After 30 min, the resultant hydrogel capsules were washed three times with 0.9% NaCl to stop cross-linking. 

##### Assay Indicators

Post induction with breast milk, the lysed solutions of the bacterial dilution ladder were used to simulate contaminated samples at the milk bank, after which a single microcapsule containing the X-Gal substrate was added to the mixture. As X-Gal diffused out of the microcapsule, it was cleaved by **β**-galactosidase, which was the basis of the colorimetric assay. 

Since the assay membrane is impermeable to **β**-galactosidase, capsule burst as a result of oncotic pressure was a binary indicator of protein expression. Due to the discrete nature of the variable (burst/no burst), the sample size needed to determine statistical power was too large. Hence, capsule burst was not used in this study as an indicator of contamination.

#### 2.2.4. Monitoring Biomolecular Activity on the Assay

##### Controls

Empty microcapsules and those containing 100 µL by volume of the X-Gal substrate (equivalent to a concentration of 2 mg/mL) were used as negative and positive controls for catalytic activity. In past studies, empty microcapsules turned blue, if alginates were not sterilized, resulting in a false positive outcome as a result of yeast contamination [9]. The controls were incubated in commercially available free β-galactosidase at a concentration of 10 mg/mL (activity: 10.3 U/mg) in order to ensure effective substrate encapsulation.

##### Categorical Scale for Optimization Runs

Biomolecular activity on the assay was monitored hourly or bi-hourly time intervals, before the optimum times for observation and data collection were executed. During optimization and confirmation experiments, the intensity (*I*) of the blue color was observed by the human eye referencing the scale presented in Figure 2, derived by assigning arbitrary values to the range of blue intensities produced in experiments. 

##### Continuous Scale for Lower LOD 

After 8 h, which was determined to be the maximum detection time to allow the color resolution at the upper end of the blue spectrum to be acceptable, a picture of the assay was taken using an iPhone camera. The image was uploaded to ImageJ version 1.51 (NIH, Bethesda, MD), and grayscale intensities were determined. Shown in Figure 3 are mean grayscale intensity (*I_G_*) values of a single well feature-extracted and imported into Excel. The intensity values were normalized (*I_N_*) according to the background intensity measured from an empty microcapsule (*I_G_*_0_) according to Equation (1): (1)$$I_{N}=\frac{I_{G-}I_{G0}}{I_{G0}}.$$
Therefore, a standard curve was generated.

#### 2.2.5. Assay Optimization 

In order to determine which variables had the greatest impact on the assay performance, an L_16_ Taguchi orthogonal array experiment was conducted with a “larger-the-better” optimization goal [32]. The experimental design matrix was conducted according to the layout in Table 2. This design entailed a four-level, five-variable array of different combinations of the following variables: volume of breast milk used for induction (*BreastMilk Volume*), lysis incubation time (*Lysing Time*), volume of X-gal substrate encapsulated in each capsule (*X-Gal Volume*, 100 µL = 2 mg/mL), concentration of *E. coli* cells (*Contamination Level*), and assay temperature (*Temperature*). Shown in Table 2 are the uncoded levels of the variables. The 16 combinations of five variables denoted as runs A–P were prepared for deposition on alginate capsules. Each run was conducted once, while subsequent confirmation runs were conducted in triplicate. 

The optimized level of parameters was determined by the difference (D), rank (R), signal-to-noise ratios (S/N), and gain (G) using the larger-the-better-optimization, which were given as following. 

Difference, *D*, was given by:(2)$$D=\sum_{ij}\frac{X}{L},$$
where *X = S/N* or *X = I* and *I* is the categorical intensity ranking based on Figure 2.

Rank, *R*, was given by: (3)$$R_{j}=Maximum\;\left(X_{ij}\right)-Minimum\;\left(X_{ij}\right),$$
where *i*, *j*, and *l* correspond to the level, factor, run#, respectively, and *L* is the total number of levels.

The theoretical signal-to-noise ratio (dB) for the *l*th experiment (*S/N)_l_* was defined by *n*, the total number of replicates per experiment, and the observed color intensity for the *m*th trial of the *l*th experiment *Y_mj_*,. Since runs were not replicated (*m* = *n* = 1), the equation for the average *S/N* was simplified as: (4)$${\left(S/N\right)}_{l}=-10\log_{10}\;\left(\frac{1}{Y_{l}^{2}}\right).$$

The gain was defined as the sum of weighted ranks (*w_j_*) for a given difference expressed in terms of signal-to-noise ratio, which was written as:(5)$$G=\sum_{j}w_{j}{\left(S/N\right)}_{ij}.$$

If a solution to the optimization problem resided in the experimental space, the optimal level of variables was a compromise between the highest gain and the acceptable response *D* expressed in terms of intensity (*I*). 

#### 2.2.6. Assay Development Material Cost 

The material cost for the individual assay was calculated by adding the amount of alginate, X-Gal, and lysing solution components.

#### 2.2.7. Statistical Analysis 

Statistical analysis was conducted using MATLAB v2019a (Mathworks, Natick, MA, USA). A one-sided student t-test for independent means at a significance level of 5% (α = 0.05) was conducted on feature-extracted color intensity measurements.

## 3. Results

### 3.1. Controls 

Shown in Figure 4a,b are the controls used for each experiment. The empty microcapsules (negative control) did not change color, while the ones containing the substrate (positive control) turned blue in the presence of commercially available free β-galactosidase, ruling out the possibility of alginate contamination. The negative controls were subsequently used for background intensity subtraction for LOD determination. Upon examination of radial cross-sections of positive controls (Figure 4b), the presence of blue color (5,5′-dibromo-4,4′-dichloro-indigo) was an indicator of uniformly dimerized X-Gal diffusion into the microcapsule. 

### 3.2. Assay Optimization and Confirmation Runs

Optimization results are summarized in Figure 5a–e and Table 3. The color changes in reaction tubes labeled as a function of experimental run are presented in Appendix A. Shown in Figure 5 is a comparison of signal-to-noise ratios plotted by individual factor after 2 and 8 h assays. There was a net quantifiable gain in dB as a result of extending the reaction time, although the trends were translatable. Intensity monitoring continued for 10 h (not shown), but the assay became saturated at the upper end of the categorical scale. The remainder of the analysis focused on a reaction time of 8 h. Based on color intensity (*I*) and signal-to-noise ratio ranks, *BreastMilk Volume* and *Contamination Level* (CFU/mL) were the top two factors, with *Temperature*, *X-Gal Volume*, and *Lysing Time* ranking third, fourth, and fifth, respectively. 

For the volume of breast milk, 150 µL of samples enabled the highest level of expression, as the S/N increased with induction volume as illustrated in Figure 5a. The activation energy of X-Gal hydrolysis and diffusion out of the assay increased with temperature. According to Figure 5e, there was a loss of 2 dB when operating at the highest level of 45 °C. A two-fold mechanism for this decrease could be proposed: (1) upon further visual examination of the samples (refer to Appendix A), this was attributed to the lack of the resolution of the blue color at the upper end of the categorical scale; and (2) although no specific studies on beta-galactosidase activity were carried out on this SCU-104 isolate, the optimum temperature for the enzyme from *E. coli* was reported to be at either 37 [33] or 50 °C [34]. Based on Figure 5e and the results obtained for the 2 h assay inspection time, the values of signal-to-noise ratios were identical at 40 and 45 °C, and thus, the heat stability of the enzyme at temperatures exceeding 40 °C may be in question for longer exposure times. With regards to bacterial concentration, the higher the level of contamination, the higher the concentration of the extracted enzyme and the theoretical X-Gal consumption. The optimum S/N was determined to be at the highest volume of 100 µL. Theoretically, the higher the *Lysing Time*, the more the enzyme extracted across combinations of *Contamination Level*. According to Figure 5b, there was an approximate loss of 0.5 dB, which according to visual inspection may be have the same root cause as operating higher temperatures or simply noise since the runs were not replicated.

The theoretical optimum levels of operation are presented in Table 3 as well as the pareto of effects expressed in terms of rank. The gain at the optimal levels of variables was calculated to be 11.78 dB.

Before proceeding to the confirmation runs results of individual runs in terms of categorical intensity (*I*) were re-examined for additional validation of theoretical analysis. Multiple optimal solutions as a combination of levels and variables were obtained as shown by the color change spectrum displayed in Appendix A, with the darkest blue intensities associated with runs G, L, M, and N, characterized by respective gains of 9.20, 10.20, 10.22, and 11.12 dB. The variable settings for run M (*I* = 8), which was chosen over run L (*I* = 8) for confirmation runs because of the higher rank of induction volume in the signal-to-noise ratio and pareto of effects. Furthermore, run N (*I* = 8) was not considered, because at 45 °C higher incubation times may affect the heat stability of the enzyme. The conditions of run G (*I* = 7.5) were not selected because of the lower gain. Hence, run M was chosen as the confirmation run.

Run M was triplicated with an average intensity score of 8 (*I* = 8), confirming the recommended conditions. The lack of resolution of the categorical scale at the upper limit did not enable the determination of a robust standard deviation.

### 3.3. LOD Determination 

In order to determine a lower LOD for the assay, optimal conditions determined for all other variables were applied. A serial dilution range of 0–10^8^ CFU/mL was subjected to a *Lysing Time* of 10 min and incubated at 40 °C with a single microcapsule containing 100 µL of X-Gal. Following feature extraction using ImageJ and intensity normalization the results in Figure 6 were generated. The results of hypothesis testing for significance between simulated levels of contamination at the 95% confidence interval (α = 0.05) are presented for average grayscale, and normalized intensities followed by subsequent integration into the respective figures. Shown in Appendix A are the samples prior to feature extraction for assay development times of 2 and 8hr as well as the grayscale and normalized intensity data.

As reflected by the norm of the calculated *p*-values in Figure 6a, there was no significant difference in grayscale intensity between 0 and 10 CFU/mL, but there was a significant difference in average intensity between 10 and 10^2^ CFU/mL (*p* = 0.015). Statistical comparisons of normalized intensity values shown in Figure 6b indicated statistically significant differences from 0 to 10^2^ CFU/mL, positioning the LOD at 10 CFU/mL. However, there was no statistical significance between 10^2^ and 10^3^ CFU/mL (*p* = 0.2078), resulting in the poor linearity of the assay when the range was extended to 10^3^ CFU/mL (R^2^ = 0.8581). Combining the analyses from both intensity scales, a binary LOD of 10^2^ CFU/mL and above was determined. 

### 3.4. Assay Development Material Cost

The cost range per assay at the optimal X-Gal concentration was calculated to be between $0.32 and $0.81. The complete assay included the gel capsule prototype ($0.0003) and X-Gal ($0.0047), in conjunction with the B-PER lysing protocol ($0.32–$0.80). 

## 4. Discussion

A unique key component of the assay was the entrapment of the X-Gal substrate in hydrogel to conserve the normal (unreacted/pre-diffused) and hydrolyzed (dimerized product diffused back into the microcapsule) conformations of X-Gal. In this case, alginate, a low-cost and well-researched biomaterial, was chosen as the entrapment gel. The maximal color change was observed after 8 h determined by multifactorial optimization. 

The composition of human milk is a dynamic variable ranging from 1 to 8 g/dL [13,35]. Sources of variations include fluctuations between and within feed periods of lactation as well as differences between mothers and populations. Since lactose and X-Gal competed as substrates for β-galactosidase, the relative concentration of each could affect assay performance. For this reason, X-Gal concentration was included as a variable in optimization experiments. The upper limit of the concentration range of the substrate of 0.2 g/dL (100 µL) used in the LOD experiments, an order of magnitude lower than the reported lactose concentrations, was chosen to maximize the resolution of the categorical scale shown in Figure 2. 

While using a single batch of bacteria and a unique sample of breast milk for LOD determination, the lack of linearity at the lower LOD can be attributed to the intensity normalization from the grayscale in the absence of a lookup table (LUT). The above root cause contributed to the variation in grayscale intensity at lower pathogen concentrations. At higher bacterial concentrations, these sources were masked by the contrast in the grayscale generated by the stronger intensity of blue color as a result of X-Gal dimerization. Combining future spectrophotometric absorbance measurements obtained from the assay supernatant and grayscale intensities will enable the correlation of pixel intensity to absorbance and hence the generation of a robust LUT. Furthermore, results will be collected at additional bacterial concentration increments in the range of 0–10^3^ CFU/mL. 

Another hypothesis could be the nonuniform radial diffusion of the dimerized X-Gal into the capsule, leading to an uneven distribution of intensity between the capsule and the solution. Although this hypothesis can be refuted at higher concentrations as illustrated in Figure 4b, it is a factor to be considered for lower detection limits. 

Coupling spectrophotometric measurements to image analysis will shed light into whether a change in immobilization morphology for X-Gal is necessary.

Based on a literature review, this proposed hydrogel-based assay has a number of competitive advantages over technology currently in use for the detection of *E. coli* and/or other pathogens. In terms of cost, the assay has the potential to become a low-cost point-of-use assay, with costs ranging from $0.32 to $0.81, depending on the concentration of reagents used for the B-Per protein extraction method. In a parallel study for the confirmation runs of the L_16_ design, the use of the Miller protocol [36] brought the total cost of the assay down significantly to $0.046. Both the Miller protocol and the B-PER protocol fall under the cost of $1.00 per assay [9]. While the Miller protocol would make the assay cheaper than the current practice of bacterial culturing, it is not environmentally friendly due to the use of chloroform. Special training for the careful handling of hazardous chloroform and access to a hood are required. The price of a product utilizing B-PER for lysis is competitive with the cost of current detection methods. The most economical immune-based test kit for water is the Watersafe^®^ Bacteria Test Kit, which detects *E. coli*, Pseudomonas, and many other forms of bacteria in 15 min at less than $3 per test [37]. 

As far as pathogen detection in biofluids across paper-based, microfluidic and electrochemical bioassays [38,39,40,41,42], the determined LOD of 10^2^ CFU/mL and above is approximately two orders of magnitude higher than the reported 0.093 CFU/mL for electrochemical immunoassays [43]. Furthermore, amplification time and associated capital costs [34,35,36,37] are not required for the proposed hydrogel assay. The closest general claims are for paper-based assays [38]. However, paper-based assays do not reach the LOD achieved by alginate assays. The optimized assay development time of eight hours falls within the range of 75 minutes [20] to 24 hours [33]. The capital costs of pasteurization to potentially eliminate *E. coli* contamination are significantly high for human breast milk banks, ranging from $35 to $81 to process approximately 100–200 ounces plus labor cost [11]. In essence, the assay outlined in this technology could be used by workers with minimal training in the milk bank. Other technologies require not only a large capital investment but specialized laboratory personnel to properly operate. 

It has been reported that *E. coli,* Staphylococcus aureus, and group B 3-haemolytic streptococci are potentially pathogenic organisms that are present in high counts in pooled human milk [44]. Practices vary between milk banks concerning milk pool volume, post-pasteurization routine bacterial assessment, bacteriological detection thresholds, and concentration standards for qualification [45,46]. However, the large majority of North American milk banks assess their donated breast milk bacteriological control based on the Human Milk Banking Association of North America (HMBANA) guideline [47], and in many countries 10^4^ CFU/mL is implemented as a cutoff number for *E. coli* and S. aureus [13]. In particular, in the province of Quebec, pre-pasteurization microbiological testing is performed for total count screening for women entering the program. The milk is tested by inoculating 100 microliters of a lot on nine blood agar plates. Plates used for the bacterial detection are incubated 48 h at 35 °C. Women having the presence of B. cereus, S. aureus, or enterobacterial at more than 10,000 CFU/mL or a total count of more than 100,000 CFU/mL have their milk rejected [12]. 

Considering the reported binary LOD cutoff of 10^2^ CFU/mL and above, this proposed hydrogel-based assay is suited to meet requirements for screening breast milk or milk for pathogenic organisms with a percentage of false positives to be determined.

## 5. Conclusions and Future Work

By simplifying the colorimetric assay design from a silicone-based paper platform to a bioactive alginate microencapsulation platform, the specific aims of sensitivity (10^5^ CFU/mL and lower), rapid detection (two–eight hours), low cost, environmental friendliness associated with the research objectives were achieved. Future efforts will encompass the following aspects: (1) examination of the effect of varying lactose concentration on induction by using various batches of breast milk; (2) use of other pathogenic bacterial strains commonly found as contaminants in breast milk; (3) optimization of lysing/extraction steps for reducing assay development time and costs, as well as increasing sensitivity; (4) development of a mobile app that can interpret and store detection results; (5) a twofold analysis of quantitative results from categorical to continuous scale with regards to image processing and automation with the aid of artificial intelligence; and (6) changes in assay morphology [48] guided by spectrophotometric analysis. 

Figure 7 is a proposed prototype for future efforts. To determine pathogen presence and quantify the concentration, 150 µL the breast milk is deposited onto a two-dimensional (2D) membrane-based sensor awaiting colorimetric detection. The paper-based alginate biosensor assay would be positioned as a point of use triage diagnostic to indicate if further testing is warranted. The cutoff for the unsafe level of contamination is shown for illustration purposes, since the level of virulence is pathogen-specific [12]; however, it satisfies the specification for milk.

## Figures and Tables

**Figure 1 sensors-20-01145-f001:**
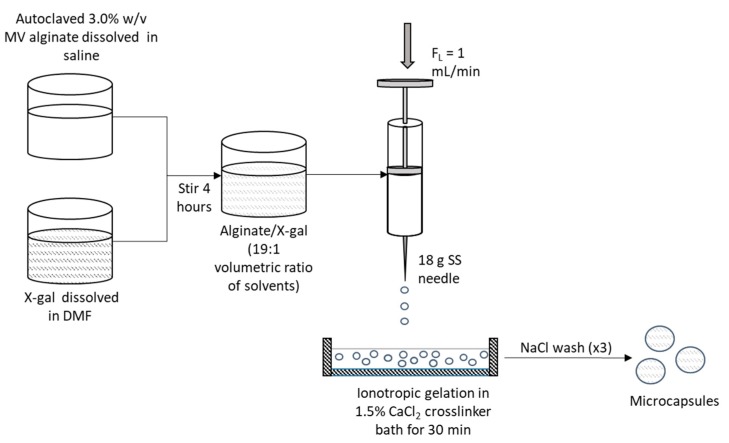
Schematic of the hydrogel capsule fabrication process.

**Figure 2 sensors-20-01145-f002:**
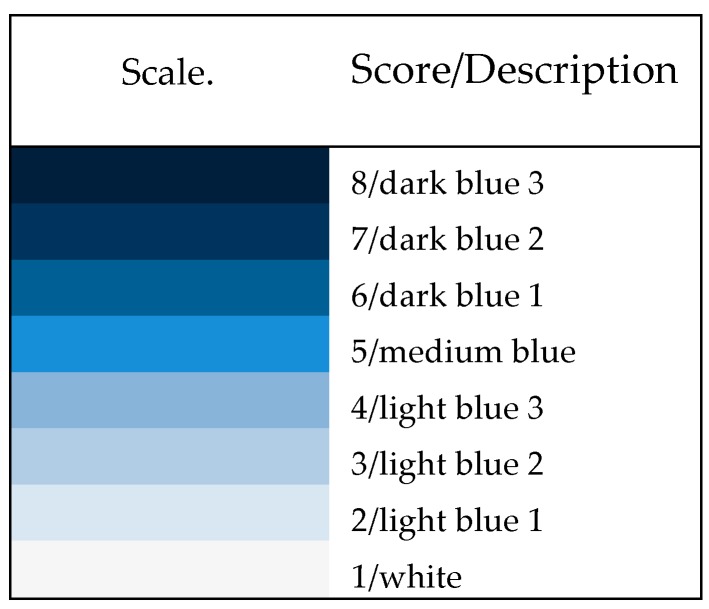
Ordinal scale for classification of catalytic activity based on visual observations.

**Figure 3 sensors-20-01145-f003:**
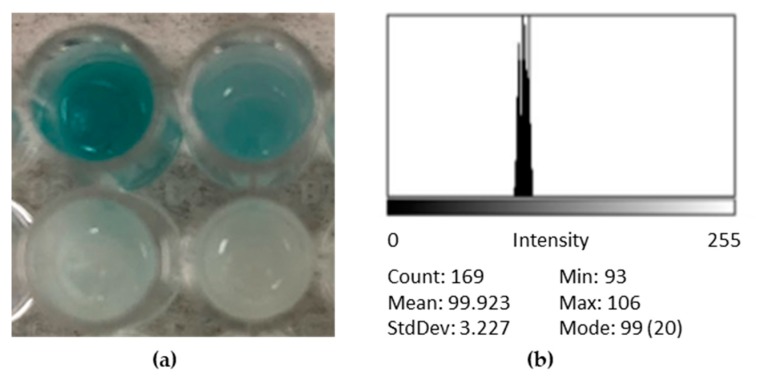
Feature-extracted grayscale intensities for recording catalytic activity using a continuous scale: (**a**) negative controls (bottom) and simulated contaminated samples (top), incubated in breast milk for 8 h; (**b**) sample grayscale intensity distribution obtained using ImageJ after cell phone image capture.

**Figure 4 sensors-20-01145-f004:**
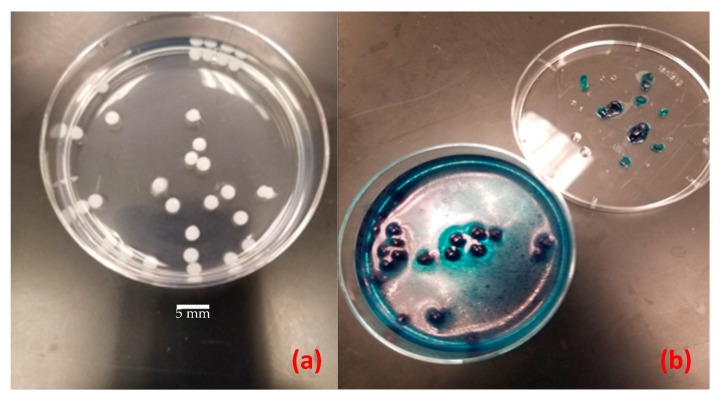
Negative (left) and positive (right) controls for the alginate bioassay-incubated controls in commercial free β-galactosidase: (**a**) empty microcapsule; (**b**) microcapsule containing 2 mg/mL of X-Gal. Scale bar represents 5 mm.

**Figure 5 sensors-20-01145-f005:**
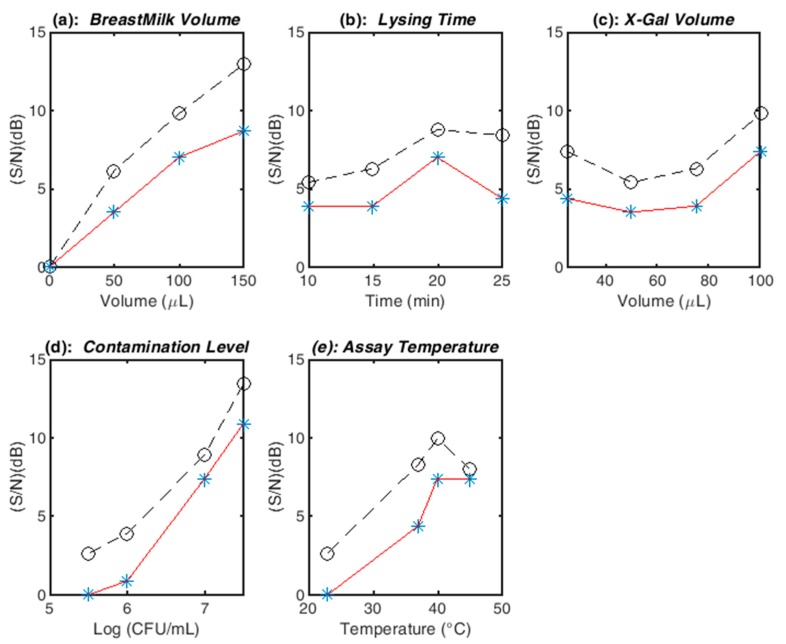
Signal-to-noise ratios plots for the Taguchi design variables: (**a**) Effect of *Breastmilk Volume*; (**b**) Effect of *Lysing Time*; (**c**) Effect of *X-Gal Volume*; (**d**) Effect of *Contamination Level*; (**e**) Effect of *Temperature*. Red and black lines correspond to incubation times of 2 h and 8 h, respectively.

**Figure 6 sensors-20-01145-f006:**
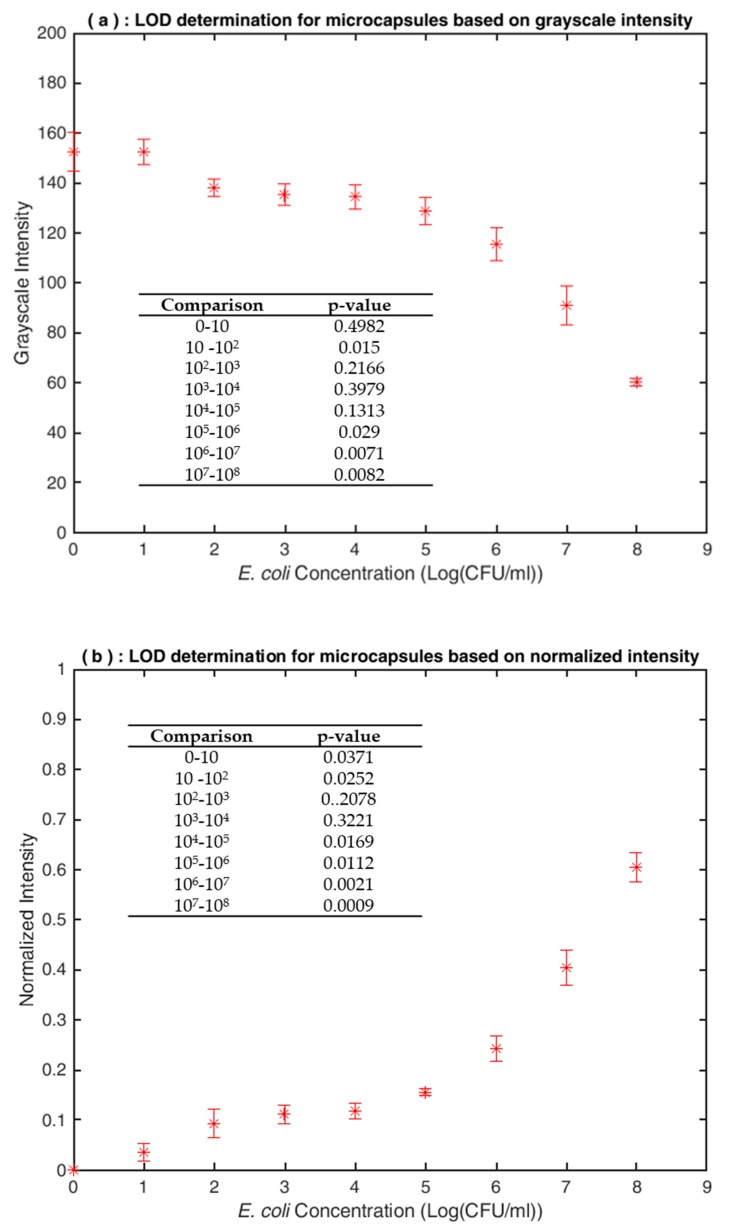
LOD determination under optimal assay conditions of run M and respective results of significance testing between simulated contamination levels at a significance level of 5% (N = 3): (**a**) (top) Grayscale intensity variation; (**b**) (bottom) Normalized intensity variation.

**Figure 7 sensors-20-01145-f007:**
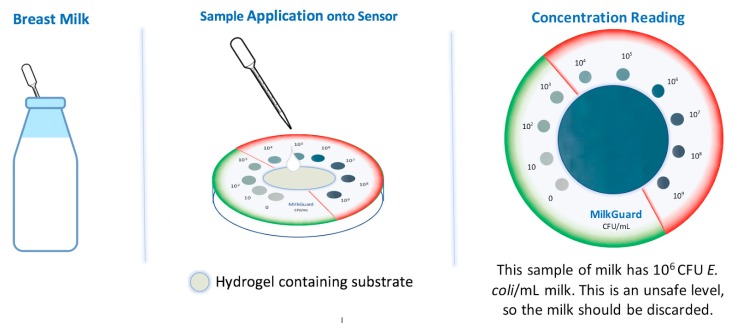
Paper-based prototype kit composed of a central alginate slab and indicator scale. Note that the cutoff of 10^4^ CFU/mL between safe and unsafe levels is established for illustration purposes.

**Table 1 sensors-20-01145-t001:** Currnt diagnostic technologies for contaminant detection in biofluids.

Product	Component Detected	Detection Method	Price	Application Setting	Time
Bacteriological testing [12,13]	Bacillus cereus, Staphylococcus aureus, and enterobacteria	Microbial culturing on agar plates	$35 to $81 to process 100–200 ounces plus labor cost [11]	Commercially available; testing samples sent to lab or testing done on site	48 h
Happy Vitals [14]	Macronutrient levels, heavy metals, vitamins, and minerals	Lab testing by microbiologists	$169.95–$695.95	Commercially available; lab screening of samples sent from mothers at home	3–5 days
Milkscreen upstring Strips [15]	Alcohol, docosahexaenoic acid	Colorimetric test strip	$14.99 for eight strips	Commercially available; testing at home	2 min
NeoGen tests for dairy products [16]	β-Lactoglobulin (BLG), casein, total milk (casein and whey proteins), and allergens	Screening microwell tests, test strips, lateral flow strips, and microwell enzyme-linked immunosorbent assay (ELISA) tests	Not advertised	Commercially available; used throughout production processes	30 min
Soleris system and vials [17]	*Escherichia coli* (*E. coli*) O157:H7	Ready-to-use vials with colorimetric indicators, incubators, and system software	Not advertised	Commercially available; used throughout production processes	4–24 h
Bioassays for quality assurance of dairy products [18]	Nutrients and pesticides	Temperature, light, and bacteria	>$1000	Laboratory research, primarily cow milk	Variable
Food bioassays [19]	General quality, carcinogen aflatoxin M1	Optical biosensing of chemiluminescence and fluorescence detection	>$1000	Laboratory research, primarily cow milk	Variable
Electrochemical DNA-based bioassay [20]	Bacillus cereus	DNA-based Au-nanoparticle-modified pencil graphite electrode (PGE)	Low	Laboratory research	Variable
Poly(methyl methacrylate) (PMMA) Lab-on-a-chip [21]	Energy content as measured by fat, protein, and lactose	Cross-flow microfiltration structure	Low	Laboratory research	Few minutes-two hours
Integrated rotary microfluidic system for point-of-care detection [22]	Salmonella Typhimurium and Vibrio parahaemolyticus	DNA extraction, loop-mediated isothermal amplification, and lateral flow strip	Low	Laboratory research	80 min
Wireless antibody-free biosensor [23]	*E. coli* C3000	Radio frequency identification (RFID)-compatible tag using gold nanoparticle markers	Low	Laboratory research	1 h

**Table 2 sensors-20-01145-t002:** Layout of experimental runs for assay optimization using the Taguchi L_16_ (5^4^) design.

Runs	*BreastMilk Volume*	*Lysing Time*	*X-Gal Volume*	*Contamination Level*	*Temperature*
A	0 µL	10 min	25 µL	3.16e^7^	23 °C
B	0 µL	15 min	50 µL	1.00e^7^	37 °C
C	0 µL	20 min	75 µL	1.00e^6^	40 °C
D	0 µL	25 min	100 µL	3.16e^5^	45 °C
E	50 µL	10 min	50 µL	1.00e^6^	45 °C
F	50 µL	15 min	25 µL	3.16e^5^	40 °C
G	50 µL	20 min	100 µL	3.16e^7^	37 °C
H	50 µL	25 min	75 µL	1.00e^7^	23 °C
I	100 µL	10 min	75 µL	3.16e^5^	37 °C
J	100 µL	15 min	100 µL	1.00e^6^	23 °C
K	100 µL	20 min	25 µL	1.00e^7^	45 °C
L	100 µL	25 min	50 µL	3.16e^7^	40 °C
M	150 µL	10 min	100 µL	1.00e^7^	40 °C
N	150 µL	15 min	75 µL	3.16e^7^	45 °C
O	150 µL	20 min	50 µL	3.16e^5^	23 °C
P	150 µL	25 min	25 µL	1.00e^6^	37 °C

**Table 3 sensors-20-01145-t003:** Optimized levels of operation for assay parameters for confirmation runs.

Variable	Rank	Optimal	Rank	Optimal	Theoretical	Confirmation
		*I*		(S/N)	Recommended	Run (M)
*BreastMilk Volume*	2	150 µL	1	150 µL	150 µL	150 µL
*Lysing Time*	5	20 min	5	20 min	20 min	10 min
*X-Gal Volume*	4	100 µL	4	100 µL	100 µL	100 µL
*Contamination Level*	1	3.16e^7^	2	3.16e^7^	1.00e^7^	1.00e^7^
*Temperature*	3	40 °C	3	40 °C	40 °C	40 °C

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
