# Peer review of "Sustainable, Alginate-Based Sensor for Detection of Escherichia coli in Human Breast Milk"

_sensors, 2020, doi:10.3390/s20041145_

Round 1

Reviewer 1 Report

I'd be interested to know what levels of pathogens are found in breastmilk to determine what an ideal sensor would achieve, and also a comment on whether specificity in terms of strains/species would be important

L263 - should increase with temperature but surely there is an optimal temperature for enzyme operation?

Do lactose levels in breastmilk vary and if so would that impact on performance of the sensor?

I'm a bit unsure about the LOD experiments and want to check I understand correctly that 150uL of breastmilk spiked at 10cfu/ml was used? So the assay can effectively detect 1.5 bacteria? I missed details of how these samples were prepared in the methods? I think this is a very interesting aspect of the study and has implications for other application areas, e.g. food or water

It would be interesting to see the time course of the reaction over the 8 hours (maybe a figure for the SI)

Author Response

I) I'd be interested to know what levels of pathogens are found in breastmilk to determine what an ideal sensor would achieve, and also a comment on whether specificity in terms of strains/species would be important

Response:

Thank you for your constructive suggestion. This paragraph has been inserted in the discussion section to address this comment,

 Practices vary between milk banks concerning milk pool volume, post-pasteurization routine bacterial assessment, bacteriological detection thresholds, and concentration standards for qualification [ref]. However, the large majority of North American milk banks assess their donated breast milk bacteriological control based on the Human Milk Banking Association of North America (HMBANA) guideline. In particular, in the province of Quebec, pre-pasteurization microbiological testing is performed for total count screening for women entering the program. The milk is tested by inoculating 100 microliters of the lot on nine blood agar plates. Plates used for the bacterial detection are incubated 48 hours at 35 celsius degree. Women having the presence of B. cereus or. S. aureus or enterobacterial at more than 10,000 CFU/ml, or a total count of more than 100,000CFU/ml have their milk rejected.              

II) )L263 - should increase with temperature but surely there is an optimal temperature for enzyme operation?

Response:

Thank you for your constructive suggestion.

The optimization section 3.2 in results has been modified to address the role of the optimum operation temperature.

“The activation energy of X-Gal hydrolysis and diffusion out of the assay should increase with temperature. According to Figure 5e there is loss of 2 dB when operating at the highest level of 45 ⁰C. A twofold mechanism for this decrease could be proposed: 1) Upon further visual examination of the samples (refer to S1), this was attributed to the lack of resolution of the blue color at the upper end of the categorical scale; and ; 2) Although no specific studies on beta-galactosidase activity have ever been carried out on this SCU-104 isolate, the optimum temperature for the enzyme from E. coli  have reported to be either at 37 ⁰C  [33] or [MM1] [MM2] 50 ⁰C [34].  Based on Figure 5e and the results of the 2 hr assay inspection time, the values of signal-to-noise ratios are identical at 40 ⁰C and 45 ⁰C and thus the heat stability of the enzyme at temperatures exceeding 40 ⁰C may be in question for longer exposure times.”

III) : Do lactose levels in breastmilk vary and if so would that impact on performance of the sensor?

Response:

Thank you for your constructive suggestion.  This has been addressed in the discussion section.

The composition of human milk is variable within feeds, diurnally, over lactation, and between mothers and populations ranging from 1-8 g/dL in average and characterized by large standard deviations [13, 35]. Any of these concentrations of lactose (> 100 mM) would be far above concentrations needed to induce ß-galactosidase expression in E. coli (< 1mM, C. Stephens, personal communication). It is possible, however, that because lactose and X-gal also compete as substrates for ß-galactosidase, that the relative concentrations of each could affect assay performance, which is why we included X-gal concentration as a variable in optimization experiments. The upper limit range of the concentration of X-gal (0.2 g/dL) used in the LOD experiments was chosen to maximize the resolution of the categorical scale shown in Figure 2.

IV) I'm a bit unsure about the LOD experiments and want to check I understand correctly that 150uL of breastmilk spiked at 10cfu/ml was used? So the assay can effectively detect 1.5 bacteria? I missed details of how these samples were prepared in the methods? I think this is a very interesting aspect of the study and has implications for other application areas, e.g. food or water.

Response:

Thank you for your constructive suggestion.

This section 3.3 has been completely revised in light of the following:

Previously pooled results from IPTG and breastmilk induction for a single starting batch of bacteria was presented making the methodology less streamlined and the mechanistic explanations convoluted, especially in terms of the built-in variable interactions in the Taguchi design. In this submission, the data presented is based SOLELY on breastmilk induction of bacteria, which represents a more realistic representation of the assay. The methodology has been changed accordingly in the cell preparation section and others to reflect this change. Regarding the lower limit of detection, the data was reanalyzed and interpreted accordingly. In the revised version, the claim to simplification of sensor design and optimization of assay conditions remain the same while a new binary LOD of 102 CFU/mL and above has been determined. In addition The sources of poor linearity have been discussed in detail with provisions to be made in future experiments.

V): It would be interesting to see the time course of the reaction over the 8 hours (maybe a figure for the SI)

Response:

Thank you for your constructive suggestion.

Supplementary material S2 has been added to address this.

Reviewer 2 Report

The manuscript "Sustainable, Alginate-Based Sensors for Detection of Escherichia Coli in Donated Human Breask Milk" by Kikuchi et al. describes an interesting and user-friendly detection method for E. Coli based the correlation of colour intensity changes on pictures taken on a smart phone.

The topic is interesting and the approach novel. However, several mandatory and major revisions are required before the manuscript is acceptable for publications in MDPI Sensors: 

The fabrication process is not well explained. Please provide more details on the fabrication process, ideally with a scheme of the fabrication steps. The text and the axis in Figure 2 are not readable, please change the figures accordingly. The optimization described in table 3 is not clear. Please explain this better, especially for what concerns the confirmation run. In particular, explain why the chosen values were selected and why the theoretical lysing time 20 min and the confirmation time 10 min.  In figure 5, it seems that the sensor curve satures at 106 CFU/ml. However then measurements were taken at 107 CFU/ml and 108 CFU/ml, a change in colour is detected. Could you please explain this, including also the incubation time? Please also comment why the linearity is soo poor, by proving more information on why this plot is useful for calibration.  In the discussion, please focus more on the discussion of the results rather than on costs. 

Author Response

I) The fabrication process is not well explained. Please provide more details on the fabrication process, ideally with a scheme of the fabrication steps.

 Response: Thank you for the constructive comment.

 Figure 1 has been created accordingly to address this shortcoming.

II) The text and the axis in Figure 2 are not readable, please change the figures accordingly.

 Response: Thank you for the constructive comment.

The figure has been revised accordingly.

III) The optimization described in table 3 is not clear. Please explain this better, especially for what concerns the confirmation run. In particular, explain why the chosen values were selected and why the theoretical lysing time 20 min and the confirmation time 10 min.

 Response: Thank you for the constructive comment.

The values of gain have been calculated for multiple conditions and the rationale for choosing the conditions of the confirmation runs have been elaborated upon.

IV) : In figure 5, it seems that the sensor curve satures at 106 CFU/ml. However then measurements were taken at 107 CFU/ml and 108 CFU/ml, a change in colour is detected. Could you please explain this, including also the incubation time? Please also comment why the linearity is soo poor, by proving more information on why this plot is useful for calibration.

Thank you for bringing up these points.

Response:

Regarding poor linearity:

This section 3.3 has been completely revised in light of the following:

Previously pooled results from IPTG and breastmilk induction for a single starting batch of bacteria was presented making the methodology less streamlined and the mechanistic explanations convoluted, especially in terms of the built-in variable interactions in the Taguchi design.  In this submission, the data presented is based SOLELY on breastmilk induction of bacteria, which represents a more realistic representation of the assay. The methodology has been changed accordingly in the cell preparation section and others to reflect this change.

Regarding the lower limit of detection, the data was reanalyzed and interpreted accordingly. In the revised version, the claim to simplification of sensor design and optimization of assay conditions remain the same while a new binary LOD of 102 CFU/mL and above has been determined. In addition, the sources of poor linearity have been discussed in detail with provisions to be made in future experiments.

2. Regarding curvature at higher concentrations::

Using a single batch of bacteria and unique sample of breast milk used for LOD determination, a source of contributing to the lack of linearity at the lower limit of detection from can be attributed to the intensity normalization from the grayscale in the absence of a lookup table (LUT). The above root cause contributes to the variation in grayscale intensity at lower pathogen concentrations. At higher bacterial concentrations, these sources are masked by the contrast in the grayscale generated by the stronger intensity of blue color as a result of X-Gal dimerization. 

V) In the discussion, please focus more on the discussion of the results rather than on costs.

 The discussion has been modified to address this shortcoming. However, since sustainability was part of the design criteria, the cost discussion needs a placeholder.

Reviewer 3 Report

Colorimetric assay for E. coli in milk, extremely long assay time of 8 hours for practical use. Only moderately interesting, major revision required as indicated below. The alginate bead filled with chromogenic enzyme substrate can not be called either sensor or biosensor. Simply, "assay" should be used in all places of the text, including title. Title, correct is "coli"; remove "donated", as this is scientifically irrelevant Abstract, do not limit statements to "breast milk" only; I am quite sure that numerous biosensors designed for just common "milk" will perform, too - include them in your introduction! L17, specify "lysis reagents". Table 1 title, "Pathogen Detection" does not correspond to most etries in the table. L88 ??? Figure 3 caption, specify the amount / activity of "free beta-galactosidase"; free means from lysate, or the commercial product? Fig. 4, x-axes titles and units should be specified. The associated discussion in the text is consequently misleading. L 285, LOD can not be "Proposed", but exactly determined. Part 3.3, blank experiment, i.e. milk sample without any spiked E. coli must be processed exactly the same way (lysis + 8 hour incubation) to determine background response - include it in Fig. 5. Otherwise any speculations on LODs are questionable. Fig. 5, BTW, you have written 10 - 10^8 CFU/ml in the text, but 0 - 8 is shown on the log scale, which corresponds to 1 - 10^8 CFU/ml. L296-299, significant difference between zero and some E. coli content should be considered. Anyway, the higher value should perhaps be considered as LOD, not the lower one. L312, x-Gal can have many conformations, but none of them is "native" as it is an artificial substance. Perhapse, the normal and hydrolysed states can be considered.

Author Response

I) Colorimetric assay for E. coli in milk, extremely long assay time of 8 hours for practical use. Only moderately interesting, major revision required as indicated below. The alginate bead filled with chromogenic enzyme substrate can not be called either sensor or biosensor. Simply, "assay" should be used in all places of the text, including title. Title, correct is "coli"; remove "donated", as this is scientifically irrelevant

Response: Thank you for the constructive suggestion as the assay needs further development before it can be called a sensor.

The word “sensor” was changed to “assay” in the title and majority of the instances. For example, for the reference to figure 7 as figure 7 represents the future development of a sensor unit. It was left sensor where the text was referring to a control volume or a physical entity where the reaction was taking place. The word “donated” was removed from its title, and we corrected Coli to coli. We apologize for the typo.

 II ) Abstract, do not limit statements to "breast milk" only; I am quite sure that numerous biosensors designed for just common "milk" will perform, too - include them in your introduction!

Response:

Thank you for your constructive suggestion. The word “breast milk” was substituted by “milk” in the first sentence. We only used “breast milk” for the description of the actual experiments.

III)  L17, specify "lysis reagents".

Response:

The authors substituted “lysis reagents” by revising to “the B-Per bacterial protein extraction kit.”

IV) : Table 1 title, "Pathogen Detection" does not correspond to most entries in the table.

Response:

The authors revised the title to “Current Solutions for Contaminant Detection in Milk and Food.”

However, culturing bacteria for quality control is an expensive (ranging from $35 to $81 to process approximately 100 – 200 ounces plus labor cost [37]) and time-consuming process (48 hours) [10] that is not strain-specific, resulting in a high incidence of false negatives. In addition, culture-based safety criteria are not standardized globally, leading to potential variation in bacterial content from different milk banks [10].  Highlighted in Table 1 are diagnostic technologies for milk and food and their varying purposes, detection methods, prices, applications, and times required to obtain results. The Happy Vitals test kit and the Milk Screen strips are commercially available, while the Biosensors for Quality Assurance of Dairy Products, Food Biosensors, and Electrochemical DNA-based biosensor, and Poly(methyl methacrylate) (PMMA) “Lab-on-a-chip” for Human Breast Milk Defatting, Integrated Rotary Microfluidic System, and Wireless Antibody-free Biosensor are still in research stage.

V) L88 ??? Figure 3 caption, specify the amount / activity of "free beta-galactosidase"; free means from lysate, or the commercial product?

Response:

Thank you for your constructive suggestion. In line 155, the concentration of the enzyme has been specified and the language was  modified for additional clarity as well as the caption for Figure 3.

VI) Fig. 4, x-axes titles and units should be specified.

Response:

Thank you for your constructive suggestion. The figures have been modified accordingly.

VIIa) The associated discussion in the text is consequently misleading. L 285, LOD can not be "Proposed", but exactly determined.

VIIb) Part 3.3, blank experiment, i.e. milk sample without any spiked E. coli must be processed exactly the same way (lysis + 8 hour incubation) to determine background response - include it in Fig. 5.  Otherwise any speculations on LODs are questionable. Fig. 5, BTW, you have written 10 - 10^8 CFU/ml in the text, but 0 - 8 is shown on the log scale, which corresponds to 1 - 10^8 CFU/ml. L296-299, significant difference between zero and some E. coli content should be considered. Anyway, the higher value should perhaps be considered as LOD, not the lower one.

Response:

Thank you for your constructive suggestion.

This section 3.3 has been completely revised in light of the following:

Previously pooled results from IPTG and breastmilk induction for a single starting batch of bacteria was presented making the methodology less streamlined and the mechanistic explanations convoluted, especially in terms of the built-in variable interactions in the Taguchi design. In this submission, the data presented is based SOLELY on breastmilk induction of bacteria, which represents a more realistic representation of the assay. The methodology has been changed accordingly in the cell preparation section and others to reflect this change. Regarding the lower limit of detection, the data was reanalyzed and interpreted accordingly. In the revised version, the claim to simplification of sensor design and optimization of assay conditions remain the same while a new binary LOD of 102 CFU/mL and above has been determined. In addition The sources of poor linearity have been discussed in detail with provisions to be made in future experiments.

VIII)  L312, x-Gal can have many conformations, but none of them is "native" as it is an artificial substance

Response:

Thank you for your constructive suggestion.

The word normal and hydrolyzed states have been used in the first paragraph of the discussion to describe the states of the entrapped X-Gal.

Round 2

Reviewer 2 Report

The manuscript has been revised sufficiently. 

Reviewer 3 Report

The revised version is ok.